# Large-Scale Survey of Missing Deciduous Anterior Teeth on Medical Examination at the Age of 3.5 Years

**DOI:** 10.3390/children9111761

**Published:** 2022-11-16

**Authors:** Tsutomu Otsuchi, Yuko Ogaya, Yuto Suehiro, Rena Okawa, Kazuhiko Nakano

**Affiliations:** 1General Incorporated Association, Dental Association of Matsubara City, Osaka 580-0044, Japan; 2Department of Pediatric Dentistry, Osaka University Graduate School of Dentistry, Osaka 565-0871, Japan

**Keywords:** missing deciduous anterior teeth, congenitally missing teeth, fused teeth, hypophosphatasia

## Abstract

Tooth anomalies in childhood may negatively affect the healthy development of the dentition and occlusion; hence, it is important to examine the actual oral condition at an early stage. The present study was performed to understand the state of missing deciduous anterior teeth in children aged 3.5 years who underwent dental checkups in Matsubara City. In total, 3508 children received oral examinations, and items such as erupted deciduous teeth and teeth anomalies were recorded. Among these children, those with missing anterior deciduous teeth were selected, and their details were analyzed. In the 216 children, there were 266 missing anterior deciduous teeth. Congenitally missing anterior deciduous teeth were observed in 80 children, and fused teeth were observed in 128 children. The missing teeth were predominantly located in the mandible and occurred more frequently on the right side. The most common reason for acquired missing teeth was trauma, and no cases of spontaneous loss due to systemic disease were found in this study. Screening for various tooth anomalies is expected to play an important role in cultivating a better understanding of the oral cavity of children, developing healthy dentitions, and contributing to the early detection of some systemic diseases.

## 1. Introduction

In Japan, health guidance and health support for mothers and children are based on the Maternal and Child Health Act, which states that its purpose is to “clarify the principles of maternal and child health, and provide health guidance, health examinations, medical care and other measures for mothers, infants and young children in order to maintain and promote the health of mothers, infants and young children, thereby contributing to the improvement of national health”. Municipal governments are obligated to provide health examinations for children aged 1.5 years and 3 years. Furthermore, dental checkups are recommended for children aged 2 years and 2.5 years as a follow-up to these health examinations and projects. Matsubara City of Osaka Prefecture has been conducting general dental checkups at the 3.5 year health examinations.

The following items are examined at dental checkups conducted by Japanese municipal governments: tooth eruption, presence or absence of dental anomalies (e.g., abnormal tooth morphology, missing teeth, or malocclusion), prevalence of dental caries, and oral soft tissue abnormalities. The dental anomalies examined in these dental checkups can be categorized into morphological abnormalities, number abnormalities, and occlusal abnormalities. The reported prevalence of morphological anomalies in the deciduous dentition, such as microdontia and fused teeth, ranges from 0.1% to 1.2% and from 0.6% to 4.1%, respectively [1,2,3,4]. The reported prevalence of number abnormalities, such as congenitally missing teeth and hyperdontia, ranges from 0.4% to 2.38% and from 0.1% to 0.8%, respectively [1,2,3,4]. The reported prevalence of occlusal abnormalities, such as anterior cross-bite and posterior cross-bite, ranges from 0.1% to 5.5% and from 0.8% to 8.7%, respectively [5]. Among these abnormalities, missing teeth is one of the most common dental anomalies that potentially affects esthetics and function. It may result in speech impairment and development of non-nutritive habits such as tongue thrusting and atypical swallowing [6,7,8,9,10].

Missing teeth can be congenital or acquired. Congenitally missing teeth (so-called hypodontia) is characterized by the developmental absence of one or more teeth [11,12]. Acquired missing teeth may occur as the result of dental trauma, neonatal tooth extraction, early childhood caries, or periodontal problems. Both congenital and acquired missing teeth can be a manifestation of systemic diseases as follows [13]. Ectodermal dysplasia is a disorder characterized by failure of development of two or more ectodermal structures involving alterations in hair, teeth, nails, and sweat glands, and hypodontia or anodontia is the oral manifestation [14]. Papillon–Lefèvre syndrome is a rare autosomal recessive disorder characterized by aggressive periodontitis, leading to the premature loss of both deciduous and permanent teeth at a very young age [15,16]. Hypophosphatasia is a rare dominant or recessive disease caused by mutations in the tissue-nonspecific alkaline phosphatase gene, and the typical oral manifestation is premature loss of deciduous teeth before 4 years of age with no root resorption [16,17,18,19].

As systemic diseases are mostly diagnosed based on the patient’s general condition, dental symptoms may be a trigger for the diagnosis when general symptoms are mild. The dental checkups conducted by the municipal governments are the first oral examination for most infants and toddlers, and it may lead them to subsequent dental examinations. Therefore, it is important to screen for various abnormalities during these dental checkups.

In the present study, the prevalence and causes of teeth abnormalities and missing teeth were investigated and analyzed at health checkups of children aged 3.5 years in Matsubara City, Osaka Prefecture, during a period of 4 years and 9 months.

## 2. Materials and Methods

### 2.1. Subjects

The protocol of the present study was approved by the Research Ethics Committee of the Japanese Society of Pediatric Dentistry (permit number 16-03). The study was performed with the cooperation of the Matsubara City Dental Association and the Dental Hygienist Association.

The participants of this study were children who underwent a dental checkup during their 3.5-year health examination in Matsubara City, Osaka Prefecture, from April 2017 to December 2021.

### 2.2. Study Protocols

A trained dentist performed all oral examinations with a mirror under bright lighting, focusing on the following items: erupted teeth, missing teeth, presence of dental caries, presence or absence of abnormal tooth morphology, dental occlusion abnormalities, oral soft tissue abnormalities, and adhesion of dental plaque. Congenitally missing teeth and fused teeth were judged as follows. Those with an insufficient number of teeth with little or no space in the area of the missing teeth were considered to have congenitally missing teeth. Particularly wide teeth or those with a vertical secant close to the center were considered to be fused teeth and were recorded as the most distal tooth (e.g., if the deciduous central incisor and deciduous lateral incisor were fused, it was recorded as the deciduous lateral incisor). Based on the results of the oral examination, children who had missing anterior deciduous teeth (including an insufficient number of teeth due to fused teeth) were selected for inclusion in the present study. After the guardians of the selected children were informed of the details of the study and had provided written informed consent, they were asked to answer a questionnaire. The questionnaire asked about the age at which they noticed the missing anterior deciduous tooth, the reason for the tooth loss, and whether they visited a dental clinic for that reason. At a later date, the health examination record of the children was investigated for details including sex, age, and presence or absence of systemic diseases or disabilities. In addition, the dental records of the selected children at their 1-year 7-month, 2-year, and 2.5-year health examinations were searched to determine when the missing anterior deciduous teeth had been recorded. For cases in which spontaneous loss of missing anterior deciduous teeth was possible, the subjects were asked to visit the dental clinic for a medical interview, oral examinations, X-ray examinations, and intraoral photography.

The prevalence of missing anterior deciduous teeth and fused teeth as shown in the oral examination records were analyzed. Statistical analyses were performed using the chi-square test. A *p* value of <0.05 was considered statistically significant.

## 3. Results

During the study period, 112 dental checkups during health examinations were conducted, and 3508 children were examined. One Down Syndrome child was excluded from the study. Four children with suspected speech delays, two with ventricular septal defects, three with intellectual disabilities, two with hearing disorders, and one with an unspecified condition were confirmed but not excluded, because these conditions were not considered to affect morphological abnormalities or the number of teeth.

Among the 3507 children, 216 (114 boys and 102 girls) had a total of 266 missing anterior deciduous teeth. Congenitally missing teeth were observed in 80 children, fused teeth were observed in 128 children, and missing teeth of other causes such as dental trauma were observed in 8 children (Figure 1). Most of the children received a dental checkup at the age of 41 or 42 months, but 12.5% of children were delayed in their health examination because we were not able to perform examinations in early 2020 because of the COVID-19 pandemic.

The missing anterior deciduous teeth were mostly located in the mandible (40 in the maxilla, 226 in the mandible) (Figure 2a). Most of the missing teeth were deciduous maxillary lateral incisors and deciduous mandibular lateral incisors, followed by deciduous mandibular canines. Missing deciduous central incisors were rarely observed in either jaw.

Most of the congenitally missing teeth were deciduous mandibular lateral incisors (76.0%), followed by deciduous maxillary lateral incisors (9.7%) and deciduous mandibular canines (8.8%) (Figure 2b). There were no congenitally missing deciduous maxillary canines on the left side, and only one on the right side. The incidence of congenitally missing teeth was significantly higher on the right side than on the left side in both the maxilla and mandible (*p* < 0.05).

Fused deciduous mandibular lateral incisors (48.3%) and fused deciduous mandibular canines (38.5%) occurred to the same extent (Figure 2c). Maxillary fused teeth of deciduous central incisors and deciduous lateral incisors were observed (13.2%); however, maxillary fused teeth of deciduous lateral incisors and deciduous canines were not recognized. The incidence of fused teeth in the mandible was higher on the right side than on the left side (*p* < 0.05).

Among the 10 missing teeth that were neither congenitally missing nor fused, two were extracted because of dental trauma or because of a natal tooth. In six cases in which the deciduous maxillary central and lateral incisors were lost, the deciduous mandibular central and lateral incisors existed without any mobility. Similarly, in one case in which the deciduous mandibular left lateral incisor was lost, the adjacent deciduous mandibular central incisor existed without any mobility. In another case in which the deciduous mandibular right central incisor was lost, the adjacent deciduous mandibular left central incisor existed without any mobility, but the details were unknown due to subject’s developmental disability. According to the child’s guardian, the tooth was already missing when they noticed and could not be found, and the child experienced constant daily trauma.

The recognized age of the missing teeth is shown in Figure 3. Most of the missing teeth were recognized at less than 2 years of age. Among them, 139 teeth were recorded as missing teeth at the 1-year 7-month health examination. Twenty teeth had no record of missing teeth for 1-year 7-month health examination for the reason unknown or since they did not received the examination. Twenty-eight teeth were recorded as teeth existing at the 1-year 7-month examination, whereas at the 2-year checkup they were either recorded as teeth missing or not received examination. Among the missing teeth that were recognized after 2 years of age, 14 teeth had no previous record because of moving from another city and the reason for the late recognition of the missing tooth was unknown. Thirty-six teeth were already recorded as missing in the 1-year 7-month health examination; hence, the examiner might have forgotten to record the results or the guardians might have forgotten the results. Four children lost a tooth or had a tooth extracted due to dental trauma, and nine children were obviously recorded mistakenly.

The reasons for the missing teeth were mostly unknown, and the children had not visited a dental clinic for that reason (Table 1). All cases that lost their teeth because of dental trauma visited a dental clinic.

## 4. Discussion

All infants and children living in Matsubara city undergo dental checkups conducted by the municipal government at the age of 1.5 years and 3.5 years. In addition, they are required to receive a dental checkup at the age of 6 years (immediately before entering elementary school). As the deciduous dentition is not yet complete at the age of 1.5 years, this is not an appropriate age to determine the presence or absence of congenitally missing deciduous anterior teeth. By contrast, all deciduous teeth are generally fully erupted at the age of 6 years [20,21]. However, the deciduous mandibular anterior teeth start to exfoliate as the permanent teeth erupt, and it is therefore difficult to determine the presence or absence of congenitally missing deciduous anterior teeth at this age. Therefore, children aged 3.5 years were selected as our study subjects because of their complete deciduous dentition but lack of anterior deciduous teeth replacement to permanent teeth.

Congenitally missing teeth constitute a prevalent multifactorial dental anomaly [11], affecting the permanent teeth more frequently than the primary dentition [22,23]. The reported frequency of congenitally missing deciduous teeth is low, being between 0.2% and 2.4% [24,25,26,27]. The reported frequency found in the present study was 2.3%, which is comparable to previous reports. The reported prevalence of congenitally missing deciduous teeth in Belgium, Turkey, India, Brazil, and New Zealand is 0.42%, 0.17–0.50%, 0.64–0.88%, 0.63%, and 0.35%, respectively [28,29,30,31,32,33,34]. These rates are lower than those in the Japanese population [12,13]. By contrast, the prevalence in the Asian countries of Taiwan and China are 1.33% to 1.80% and 4.06%, respectively (i.e., approximately the same as or higher than those in Japan) [35,36,37]. Thus, although the prevalence of congenitally missing teeth differs among continents and races, these variations are unlikely to change over time [27]. In the present study, most missing teeth were observed in deciduous mandibular lateral incisors, which was consistent with previous reports [25,26,38,39]. However, we found that the deciduous mandibular canine was the second most common congenitally missing tooth; this finding differed from some reports that found no missing deciduous canines [25,39].

Fused teeth are believed to be related to physical force or pressure from the follicles of adjacent teeth, hereditary conditions, and racial determinants, although the exact etiology is unknown [24]. Some researchers have reported that the incidence varies according to race. The prevalence of fused teeth in Japanese children was reported to be significantly higher than that found in studies of American and Scandinavian children [38]. In addition, Asian and Asian-derived populations show a range of prevalence of fused teeth from 1.2% to 5.2%, while European and European-derived populations universally exhibit frequencies of less than 1.0% [40]. Many researchers have reported the frequency of fused teeth in the Japanese population, but most of these studies were calculated with data from dental examinations at dental clinics, kindergartens, and nursery schools [39]. Several papers reported that the prevalence of fused teeth in infant health examinations targeting almost all the relevant age groups in a certain area was 4.5%, 3.1%, and 4.10%, which is relatively consistent with the results of the present study (3.6%) [12,25,26]. Furthermore, they reported that fused anterior deciduous teeth were predominantly located in the mandible rather than the maxilla, and that mandibular lateral incisors were often missing. They also occurred more often on the right side than the left side [25,26]. These results did not differ from those of the present study.

The frequency of occurrence of congenitally missing and fused deciduous teeth during the 7-year period from 1983 to 1989 ranged from 4.0% to 5.0%, indicating an increase compared with a previous study dating from the 1950s [26]. In the present study, the frequency was 6.2%, suggesting a further increase.

When designing a study of missing anterior deciduous teeth, a cross-sectional survey based solely on the results from the 3.5-year dental health examination will only reveal the frequency of occurrence at that particular age. Therefore, to find out when and why anterior deciduous teeth are missing, it is important to retrospectively examine when the missing teeth were recorded in the children’s previous dental health examinations. In addition, it is important to interview guardians about when they realized the tooth was missing and how it was lost. In the present study, by taking this additional information into consideration, we were able to determine more accurately whether the missing teeth were congenitally missing or spontaneously lost.

Dental checkups during health examinations by the municipal government are performed in a room at the health center. Although a visual intraoral examination is performed, an X-ray examinations is not. Therefore, the diagnoses of congenitally missing teeth and fused teeth are based on visual examination of the oral cavity. This is a limitation of the present study. When absent deciduous teeth are found during the intraoral examination, the possibility of delayed tooth eruption (as opposed to congenitally missing teeth) can be considered. However, failure of primary tooth eruption is reportedly very rare; the prevalence of impacted primary teeth is 1:10,000, and it usually involves the primary second molars [41,42]. Although radiographic examination is essential for a definitive diagnosis, as a screening for missing teeth, we believe that we have still provided meaningful data within this limitation.

It is considered relatively easy for a dentist to determine that an anterior deciduous tooth is missing at a 3.5-year health examination. However, even if the dentist notices it, it is often overlooked without being considered a serious matter. This is because, except in confirmed cases (from interviewing the guardians) of teeth disintegrating and being extracted due to dental caries or dental trauma, the reasons for missing teeth are mostly indeterminable, and the teeth will be judged as being congenitally missing or fused. However, attention should be paid to cases in which the probability of spontaneous tooth loss cannot be denied. HPP is a systemic disease in which early exfoliation of deciduous teeth, mostly the mandibular central incisors, before 4 years of age with no root resorption, is the typical oral manifestation [19,20,21]. It is characterized by bone hypomineralization and/or early exfoliation of deciduous teeth [19,20,21]. Mild types of HPP are sometimes overlooked due to the absence of life-threatening bone symptoms [43,44,45]. However, HPP is a progressive disease, with the possibility that some patients exhibit odonto-HPP with only dental manifestations at the time of diagnosis, and who subsequently develop childhood or adult HPP with bone symptoms [21,44]. Therefore, early diagnosis of odonto-HPP is important. There are some reports that dental manifestations such as early exfoliation of deciduous teeth has led to the diagnosis of HPP [45,46]. Hence, it is important to pay attention to early exfoliation of deciduous mandibular anterior teeth in the dental health checkup conducted by the government in Japan under the age of 4 years, as it may contribute to early detection of HPP. In the present study, although eight teeth were exfoliated due to trauma, no deciduous mandibular central incisors showed spontaneous loss or mobility, indicating that none of these cases were suspected HPP. The prevalence of mild type HPP is reportedly approximately 1 in 6300 people [47]; therefore, this finding is considered to be reasonable. As the number of subjects increase, some suspicious cases may emerge in the future.

Dentists should pay more attention to missing deciduous anterior teeth at dental health checkups. In fact, an increasing number of local governments have set new check items for deciduous teeth mobility and early exfoliation in local dental health checkups conducted at an early age. Based on the results of the present study, fused teeth, congenitally missing teeth, early exfoliation, and mobile teeth were added to the check items for infant dental examinations in Matsubara City, Osaka Prefecture, from April 2022. We hope that this will contribute to the healthy growth of the deciduous dentition and assist in early diagnosis of HPP.

## 5. Conclusions

In summary, we investigated missing anterior deciduous teeth in the 3.5-year medical examination for a period of 4 years and 9 months. Through the investigation, missing anterior deciduous teeth were predominantly found in the mandible, mostly in the deciduous mandibular lateral incisors, and more often on the right side than the left side. Acquired missing teeth were mostly a result of dental trauma, and no cases of spontaneous loss due to suspected systemic diseases were found in this study. Screening for various tooth abnormalities is expected to play an important role in cultivating a better understanding of the oral cavity of children, developing healthy dentitions, and contributing to the early detection of some systemic diseases.

## Figures and Tables

**Figure 1 children-09-01761-f001:**
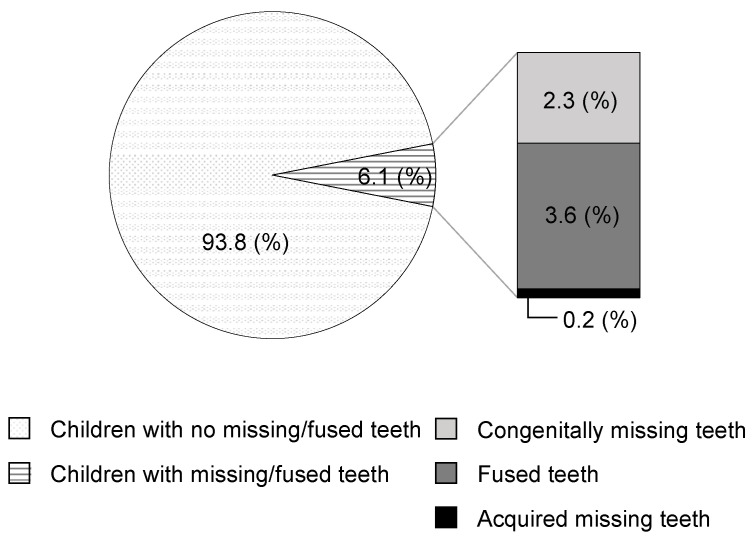
Prevalence of missing teeth and fused teeth among children in this study.

**Figure 2 children-09-01761-f002:**
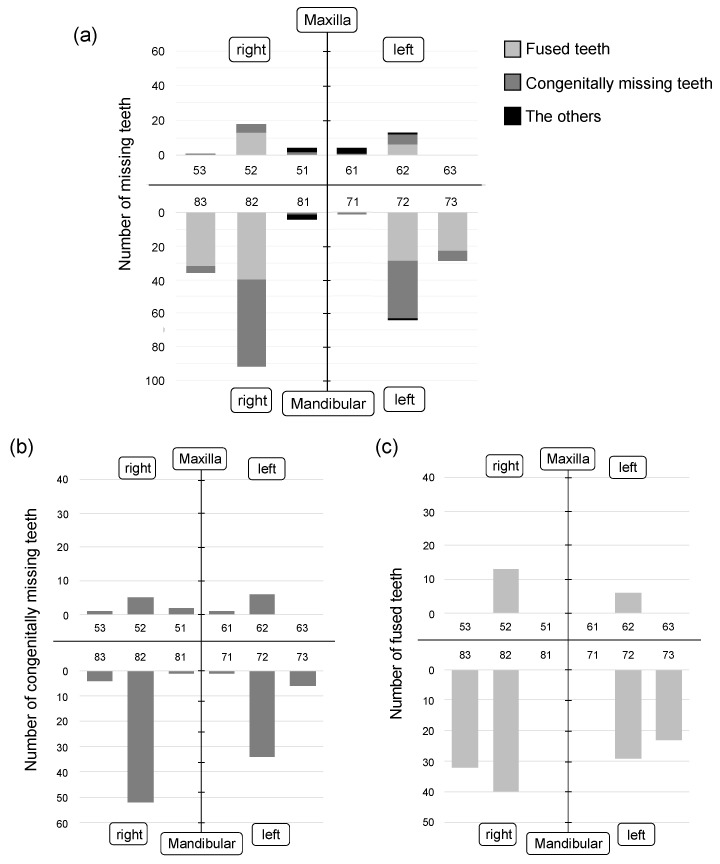
Location and number of missing deciduous teeth. (**a**) Location and number of all missing deciduous teeth. (**b**) Location and number of congenitally missing deciduous teeth. (**c**) Location and number of fused deciduous teeth. Most of the missing anterior deciduous teeth were located in the mandible.

**Figure 3 children-09-01761-f003:**
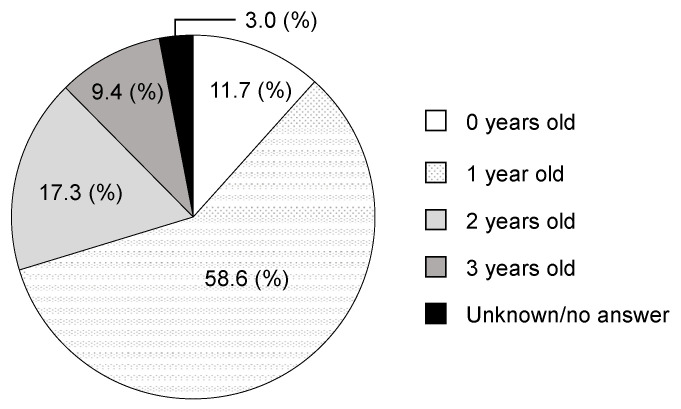
Age at which missing teeth were recognized. Most of the missing teeth were recognized before the age of 2 years.

**Table 1 children-09-01761-t001:** Summary of the questionnaire survey.

Reason for Missing Teeth	Number of Teeth (%)
Avulsion or exfoliation due to dental trauma	11 (4.1)
Extraction due to dental trauma data	1 (0.4)
Extraction due to dental caries	1 (0.4)
Missing due to cleft lip and palate	1 (0.4)
Unknown	229 (86.1)
No answer	23 (8.6)

## Data Availability

Data are contained within the article.

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
