# Peer review of "Large-Scale Survey of Missing Deciduous Anterior Teeth on Medical Examination at the Age of 3.5 Years"

_children, 2022, doi:10.3390/children9111761_

Round 1

Reviewer 1 Report

Dear authors, 

Congratulations on the job you have done and presented in this manuscript. I believe that your job can be published after some minor corrections in the introduction and material and method section. Please see the attachment.

Author Response

Response to reviewers

Reviewer 1

Abstract:

Please state the aim of the study

(Response) We have added the aim of the study that to understand the state of missing deciduous anterior teeth in children (Line 9).

Please try to add more details regarding methodology conducted, what do you mean by medical health examination? Were there any tooth indexes analyzed and included in the methodology?

(Response) Thank you for your kind suggestion. Due to the 200 word limit of abstract, we tried to add methodology and recorded items as much as we could (Line 11).

Medical health examination is what Matsubara city conducts for 3.5 years and oral examination are performed at the same day. Since the expression was unclear, we revised the sentences (Line 12).

Introduction:

This section is clearly presented and in my opinion meets all the required standards.

However I would recommend expanding this section with a paragraph or two stating very recent, state-of-the-art information related to the subject. Also try adding some recent references, as it stands now is not good enough. Ref.3 is very old and there are only 4 references in the entire section.

(Response) Thank you for your kind suggestion. We have added paragraph by adding information about the dental anomalies and data of systemic diseases associated with missing teeth. We have added reference of 2022 and totally 19 references in the entire section.

Materials and Methods:

In this section please state the exact protocol followed during examination, who performed the medical examination, instruments used…

(Response) We have added the protocol followed during examination as a trained dentist performed an oral examination with a mirror under bright lighting (Line 80).

How was the retrospective analysis performed? Medical papers were used? All these details are required

(Response) Thank you for pointing it out. We have mistranslated the explanation. We have revised the sentences that we looked back the past examination records of the each subjects to compare the record with the present records (Line 97).

This line is a little bit unclear. Please state what do you mean by 112 health examinations?

(Response) Thank you for your advice, we have revised the sentence to 112 times of dental checkup at the health examinations (Line 103).

I would suggest the authors to add a graph or image regarding this data, it is very unclear for the general reader.

(Response) Thank you for your kind suggestion, we have added figure 1 that represent prevalence of missing teeth and fused teeth among study subjects.

Please state what are the other reasons

(Response) We have explain the other reasons such as dental trauma (Line 111).

There has to be an error here, in line 86 you state that there were 8 missing teeth, now there are 10

(Response) Thank you for your pointing it out. The number that we state in line 86 was children who had missing teeth and in line 111 was exact number of missing teeth.

Reviewer 2 Report

Dear authors,

Thank you for submitting the article entitled “Large-scale survey of missing deciduous anterior teeth on medical examination at the age of 3.5 years”. My feedback is below:

Introduction section:

-“You mention that missing teeth is one of the most common dental anomalies”, please provide data/percentage for that statement.

-“Missing teeth associated with systemic diseases”, please develop this sentence further… provide more data/percentage, applicable worldwide and/or Japan ?

- Introduction is very short, only two paragraphs, you should develop it more.

Materials and Methods section:

-You only considered the space to believe there were missing teeth. Did you verify it with another method such as radiographs? What if patient had delay eruption instead of missing tooth?

Discussion section:

Please describe why you selected the age of 3.5 for the children in this study.

-“The prevalence of missing teeth in the Japanese population was shown to be higher than other countries in several reports…” please provide more details/numbers/name of countries for that comparison.

Thanks.

Author Response

Reviewer 2

Introduction:

“You mention that missing teeth is one of the most common dental anomalies”, provide data/percentage for that statement.

(Response) We have added data as missing teeth being one of the most common dental anomalies (Line 41).

“Missing teeth associated with systemic disease”, develop this sentence further and provide more data/percentage, applicable worldwide and/or Japan.

(Response) We have added data of missing teeth association with systemic diseases with further explanation of term missing teeth (Line 49).

Introduction is very short, only two paragraphs, you should develop it more.

(Response) In accordance with reviewer’s suggestion, we have developed sentences of introduction and revised to four paragraphs.

Materials and Methods:

You only considered the space to believe there were missing teeth. Did you verify it with another method such as radiographs? What if patient had delay eruption instead of missing tooth?

(Response) Thank you for pointing it out. We have described the limitation of the present study with a reference of delayed eruption of deciduous teeth (Line 227).

Discussion:

Describe why you selected the age of 3.5 for the children in this study.

(Response) We have described why we selected the age of 3.5 for the children in this study that the children aged 3.5 years generally have complete deciduous dentition but have no exfoliation in anterior deciduous teeth due to the eruption of permanent teeth (Line 171).

“The prevalence of missing teeth in the Japanese population was shown to be higher than other countries in several reports…” provide more details/number/name of countries for that comparison.

(Response) We have added more details, number and name of countries for the comparison (Line 186).